# The Effect of Exogenous Oxytetracycline on High-Temperature Anaerobic Digestion of Elements in Swine Wastewater

**Zhongda Hu, Zijing Fan, Qixuan Song** 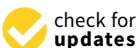**, Rabia Khatoon, Mei Zhang, Ning Wang and Xingzhang Luo ***

Department of Environmental Science and Engineering, Fudan University, Shanghai 200433, China;
17210740025@fudan.edu.cn (Z.H.); 18210740041@fudan.edu.cn (Z.F.); 17110740032@fudan.edu.cn (Q.S.);
17110740061@fudan.edu.cn (R.K.); 18210740070@fudan.edu.cn (M.Z.); 17210740032@fudan.edu.cn (N.W.)
* Correspondence: lxz@fudan.edu.cn; Tel.: +86-186-2108-0212

**Abstract:** Tetracycline antibiotics (TCs) are a common type of antibiotic found in swine wastewater. Oxytetracycline (OTC) is a significant type of TC. This study mainly examined the influence of OTC on high-temperature anaerobic digestion by adding OTC to collections of swine wastewater at different times during the digestion process. The results showed that high-temperature anaerobic digestion was suitable for the removal of TCs, with an 87% OTC removal efficiency achieved by day 20. Additionally, OTC added from external sources was found to inhibit the chlortetracycline degradation process and affect the first-order degradation kinetic model of TCs. Complexation reactions were the main ways in which OTC affected the heavy metal content of the water. The exogenous addition of OTC was found to inhibit the activity of some digester microbial strains, reduce the proportion of dominant strains, such as *MBA03*, and kill certain specific strains. This performance alteration was most obvious when OTC was added in the middle of the reaction.

**Keywords:** swine wastewater; high-temperature anaerobic digestion; tetracycline antibiotics; oxytetracycline

## 1. Introduction

In 2018, there were 790.5 million pigs in the whole world, and global pork production was about 118.8 million tons [1]. Swine wastewater has been one of the most widely distributed pollution sources globally, and the discharge of pig wastewater has increased gradually over recent years. Furthermore, China is one of the countries with the highest pork production, with pork accounting for 65% of all meat consumed in China [2]. A previous study estimated that each pig farm in China produces approximately 1300 m$^3$ of pig wastewater each year [3]. In South Korea, swine wastewater accounted for approximately 53% on average of all livestock excreta [4].

Antibiotic pollution has become an increasingly serious concern in the past few decades, increasingly attracting attention. Antibiotics are also a non-negligible part of swine wastewater, and swine wastewater is known to be a major source of environmental antibiotic input. The livestock and poultry industries consume more than 84,000 tons of veterinary antibiotics every year in China, and about 58% of these are directly or indirectly discharged into the natural environment [5]. It is estimated that from 2010 to 2030, global antibiotic use will increase by 67%. In Brazil, Russia, India, China, and South Africa, the rate of usage is expected to double [6].

Swine wastewater contains various types of antibiotics, including tetracyclines, sulfonamides, macrolides, fluoroquinolones, and other veterinary antibiotics [7]. Tetracyclines are the most common veterinary antibiotics in pig fodder, as well as the most threatening antibiotics to the natural environment, with concentrations as high as 685.60 µg/L being used [8]. An American Food and Drug Administration report showed that 80% of antibiotics sold in the United States are used for livestock production. The ways that different types of antibiotics were used in livestock production in 2017 are shown in Figure 1.

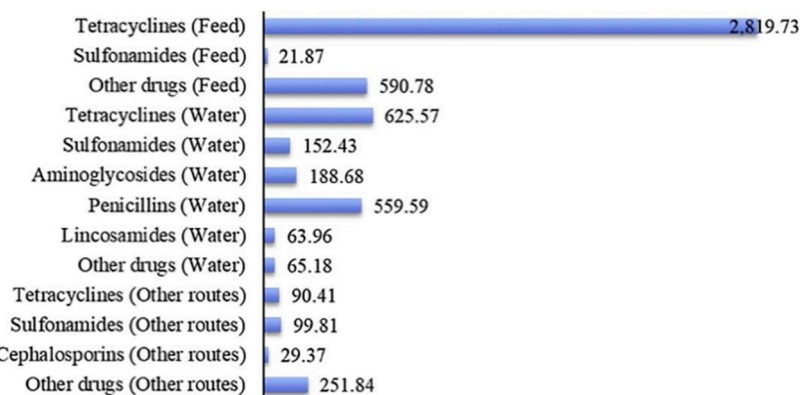

**Figure 1.** The amount and route of different classes of antibiotics used in livestock in the United States in 2017, in 1000 kg (F. D. Administration, 2017).

Antibiotics are difficult to degrade in the natural environment and can have a terrible impact on soil and water. Antibiotic administration has direct effects on the enteric and pathogenic bacteria in livestock animals, and environmental bacteria are also affected if antibiotic residues and metabolites are released into the environment [9]. Currently, antibiotic-resistant bacteria with resistance to sulfonamides and tetracyclines are detected ubiquitously in Asia [10].

This requires pig farms to treat swine wastewater to remove residual antibiotics before it is discharged; however, conventional treatment technology only shows good performance in the removal of general polluting factors, such as chemical oxygen demand (COD) and total nitrogen (TN). A study reporting on sewage treatment plants in the Jiulongjiang Basin found that the average removal efficiency of TCs in sewage treatment plants is mainly based on anaerobic–anoxic–oxic processes (A2/O), and adsorption biodegradation (A/B) was between −71.6% and 56.3% [11]. Zhang et al.'s research found that the removal rate of TCs by composting was less than 63.7%, but that the composting cycle was 171 days [12]. Glutathione S-transferase (GST) is a type of protein that can be used to specifically degrade refractory and impediment antibiotics; however, the degradation efficiency of this method for TCs was found to be only 30% [13]. On account of this, there is an urgent need to find a swine wastewater treatment process that can efficiently degrade TCs.

Anaerobic digestion is a common way to degrade swine wastewater, as it has a high degradation efficiency, low energy consumption, and is easy to carry out. Anaerobic digestion has also been shown to have an excellent performance when it comes to the degradation of TCs. Researchers evaluated the impact of different concentrations of tetracycline on the performance of anaerobic treatment and found no major sustained impact on methane production, demonstrating that anaerobic digestion was stable and reliable in the presence of tetracycline [14]. During the anaerobic digestion of manure from medicated calves, the chlortetracycline (CTC) concentration decreased approximately 75% during the 33-day digestion period [15]. Wang et al.'s results showed that the removal rate of TCs in the supernatant could be up to 90%–100% during swine manure anaerobic digestion. However, their removal rates in the solid phase were only around 40% [16]. The OTC removal efficiency in a different thermophilic composting study was found to be over 90% [17].

Ma et al.'s study indicated that thermophilic anaerobic digesters more effectively reduced concentrations of TCs compared to mesophilic digesters [18]. During anaerobic digestion, the number of antibiotic resistance genes decreased sharply with increasing temperature [19]. However, the removal rate did not increase with increasing temperature. Despite this, some studies have shown that when the anaerobic digestion temperature exceeds 60 °C, the reduction of antibiotic resistance genes is not as high as when the temperature is below 55 °C [20].

OTC is a crucial broad-spectrum antibiotic [21] and can reliably inhibit a variety of bacteria at low cost [22]. OTC has been used widely in pig farms to control diseases caused by various bacteria [23]. However, there are few studies looking at how changes in OTC content affect the difficulty of removing pollutants from swine wastewater during anaerobic digestion.

The most common and effective method used to deal with swine wastewater and degrade antibiotics has been high-temperature digestion. As the content of antibiotics varies in different collections of swine wastewater, there are few studies aiming to examine the effect of antibiotic content on high-temperature swine wastewater digestion and how various components in the wastewater affect each other in the process. This study focused on the effect of OTC addition in high-temperature anaerobic digestion (55 °C) and the interactions between TCs, heavy metals, and the microbial community, aiming at providing a theoretical basis for the study of high-temperature anaerobic digestion of TCs.

## 2. Materials and Methods

### 2.1. Swine Wastewater and Granular Sludge

The swine wastewater used in this study was taken from a pig farm in Jiaxing City, Zhejiang Province, China. It was fresh pig manure. The advantage of fresh pig manure was that it could to avoid the degradation of antibiotics due to time and temperature. The solid content of pig manure was about 5%, and the basic indicators, such as pH, total organic carbon (TOC), total phosphorus (TP), and total nitrogen (TN), are shown in Table 1. The basic indicators were tested after mixing, centrifuging, and diluting. TOC and TN were detected by a TOC analyzer, and TP was analyzed manually by the molybdate calibration method.

**Table 1.** Basic indicators of swine wastewater.

| Indicator | Value (Unit) |
|:---:|:---:|
| pH | $8.01 \pm 0.1$ |
| TOC | $2598.8 \pm 130.0$ mg/L |
| TN | $1374.0 \pm 54.4$ mg/L |
| TP | $112.8 \pm 5.3$ mg/L |

The inoculated sludge in this study was granular sludge obtained from the Shanghai Songjiang Tsingtao Brewery and was a black granular solid with many extracellular polymers. Solid and liquid reagents were stored at 4 °C, and ordinary sampling reagents were stored at −20 °C.

### 2.2. Experimental Design

High-temperature anaerobic digestion was carried out in a digestion reactor, with a water bath maintaining the temperature in the reactor at 55 °C. For the uniformity and completeness of the reaction process, each digestion reactor was equipped with a stirring device, which could constantly stir with 300 r/min. The reactor provided a closed anaerobic environment for the digestion process.

The main experimental process was carried out in four groups of anaerobic digestion reactors, each with a capacity of 1000 mL. Each reactor was supplemented with a total of 1000 mL of swine wastewater and 50 g of solidified granular sludge. Before digestion, all reactors were infused with nitrogen for 30 min to obtain an anaerobic environment. Reactors were then subjected to high-temperature anaerobic digestion at 55 °C. After the digestion was started, groups 2, 3 and 4 were spiked with 1 μg of OTC by different methods. More details are shown in Table 2.

**Table 2.** Time of addition and quantity of OTC.

| Group | 0 | 2 | 4 | 6 | 8 | 10 | 12 | 14 | 16 | 18 | 20 |
|-------|------|---|------|---|------|----|----|----|----|----|----|
| 1 | 0 | 0 | 0 | 0 | 0 | 0 | 0 | 0 | 0 | 0 | 0 |
| 2 | 1.00 | 0 | 0 | 0 | 0 | 0 | 0 | 0 | 0 | 0 | 0 |
| 3 | 0.33 | 0 | 0.33 | 0 | 0.33 | 0 | 0 | 0 | 0 | 0 | 0 |
| 4 | 0 | 0 | 1.00 | 0 | 0 | 0 | 0 | 0 | 0 | 0 | 0 |

The high-temperature anaerobic digestion process took 40–60 days to optimize; however, the core anaerobic digestion process mainly occurred in the first 20 days. With the focus being on the process of high-temperature anaerobic degradation of TCs and the effects of OTC addition on this process, the main experimental study took place over the first 20 days. During the reaction process, 12 samples were taken to do further analysis at 0, 0.5, 1, 2, 3, 4, 5, 6, 8, 12, 16, and 19 days. A total of 5 mL of mixture was taken in each sampling by syringe, which aimed at keeping oxygen from entering the reactors. At the end of each sampling, reactors were refilled with nitrogen to ensure an anaerobic environment. The 5 mL mixture was used to detect basic indicators, TCs, microorganisms, and heavy metals.

Water indicators showed the chemical changes and treatment effects during high-temperature anaerobic digestion, including TOC, TN, and TP, which were obtained using a TOC analyzer and chemical detection methods. Heavy metal content (Cu, Zn, and Fe) was detected by a heavy metal analyzer after microwave digestion with strong acid to display the migration and transformation of heavy metals during the digestion process. Microbial communities were detected by Majorbio to analyze community changes during the digestion process. After solid phase extraction, the concentrations of antibiotic contents, including tetracycline (TC), oxytetracycline (OTC), chlortetracycline (CTC), and doxycycline (DOC), were detected by liquid chromatography and mass spectrometry so that the degradation effect from high-temperature anaerobic digestion could be explored.

### 2.3. Analysis Instruments and Methods

#### 2.3.1. Instruments

Ultra-performance liquid chromatography (UPLC) was used in the analysis, and the mass spectrometer employed was a Xevo TQ-S (Waters, Milford, MA, USA). The TOC and TN were detected by TOC-L CPH (Shimadzu, Kyoto, Japan). The heavy metals were determined using a ContrAA 300 (Jena, Germany). The water used in this study was treated with Milli-Q when needed for sample analysis.

#### 2.3.2. Solid Phase Extraction

The four major tetracycline antibiotics in this study, including TC, OTC, CTC, and DOC, were purchased from the Aladdin Company. Methanol and formic acid (HPLC 99.9%) used in sample detection were purchased from J&K Scientific, and the solid phase extraction column used in the solid phase extraction process was purchased from Waters.

Antibiotic contents of swine wastewater were extracted by solid phase extraction. First, swine wastewater was centrifuged at 10,000 r/min for 10 min, and 5 mL supernatant was taken and diluted 10-fold with ultrapure water. For storage, 0.2 g of EDTA-Na$_2$ was added before storage at 4 °C in darkness. The obtained samples were treated within 3 days. The water samples needed to be filtered through a 0.45 μm glass fiber filter membrane, and pH was adjusted to 2.5–4 using 6 mol/L hydrochloric acid. After adjustment, HLB solid-phase extraction cartridges were activated with a triple rinse of a 2 mL methanol, 2 mL water, and 2 mL hydrochloric acid solution with pH = 3. After passing the water sample through the solid-phase extraction instrument, the instrument was pumped to dry at 5 mL/min, and continued to be vacuum dried for 30 min. The small column was rinsed with 2 mL of 5% methanol, before slowly eluting with 6 mL of methanol solution, which was collected with a 10 mL centrifuge tube. Nitrogen was blown until the sample was near dryness, and the sample was topped up to 1 mL with methanol, filtered with a 0.22 μm

organic filter, and transferred to a 2 mL brown sample bottle which was protected from light.

### 2.3.3. Liquid Chromatography Conditions

The UPLC used in this experiment was a quaternary pump system with a vacuum degasser, an autosampler, and a constant temperature column box. The chromatography column was a Waters ACQUITY UPLC BEH C18 with a particle size of 1.7 μm. During the detection process, the temperatures of the sample chamber of the liquid chromatograph and column were 25 °C and 30 °C, respectively. The sample flow rate was 0.3 mL/min, and the injection volume was 20 μL per needle. The experimental analysis using a binary solvent system included mobile phase A (0.1% formic acid in water) and phase B (methanol). The gradient concentration of the mobile phase changed within the first minute, using 90% A and 10% B. Between 1 and 1.5 min, the mobile phase was changed linearly to 10% A and 90% B, which was maintained from 1.5–4 min. After that, from 4–4.2 min, the gradient of the mobile phase was changed back to the initial mix and was maintained until the end of 5 min.

### 2.3.4. Mass Spectrometry Conditions

A three-stage quadrupole mass spectrometer was used for the mass spectrometry analysis, with an accuracy reaching one part per trillion (ppt) levels. The mass spectrometer was equipped with an ESI source and Masslynex software was used for instrument control, data acquisition, and evaluation. The gas flow was transferred from the LC column to the MS using the two strongest and most specific fragment ions to determine the multi-residue analysis. The entire process was analyzed in cation mode, with a scan time of 0.5 s, a scan width of 0.2, argon gas used as the collision gas, a flow rate of 0.15 mL/min, a spray voltage of 3.5 kV, a cone voltage of 75 V, a desolvation gas temperature of 400 °C, a desolvation gas flow rate of 800 L/h, and a cone gas flow rate of 150 L/h. Table 3 and Figure 2 show the analytic parameters and the oxytetracycline mass spectrogram, respectively.

**Table 3.** Main parameters for mass spectrometry analysis.

| Compound | Retention Time (min) | Formula | Parent (*m/z*) | Daughters (*m/z*) | Cone Voltage (V) | Collision Energy (eV) |
|---|---|---|---|---|---|---|
| OTC | 2.34 | $C_{22}H_{24}N_2O_8$ | 444.45 | 409.97 | 32 | 28 |
| TC | 2.35 | $C_{22}H_{24}N_2O_9$ | 460.45 | 426.00 | 32 | 18 |
| CTC | 2.47 | $C_{22}H_{23}ClN_2O_8$ | 479.02 | 443.98 | 40 | 30 |
| DOX | 2.52 | $C_{22}H_{24}N_2O_8$ | 445.06 | 153.96 | 40 | 30 |

### 2.3.5. Standard Curve

The sample concentrations were calculated by the external standard method based on the peak area of the product ions monitored. The chromatographically pure analyte was dissolved in methanol to prepare a 100 mg/L stock solution and stored in an amber vial at 4 °C in darkness. The standard curves of the four analytes were diluted to 0.1, 1.0, 2.0, 5.0, 10.0, 20.0, 50.0, and 100.0 μg/L in the initial concentration mobile phase mixture. The standard curve was used within 2 h each time. The $R^2$ of the standard curve used in this experiment was above 0.9999.

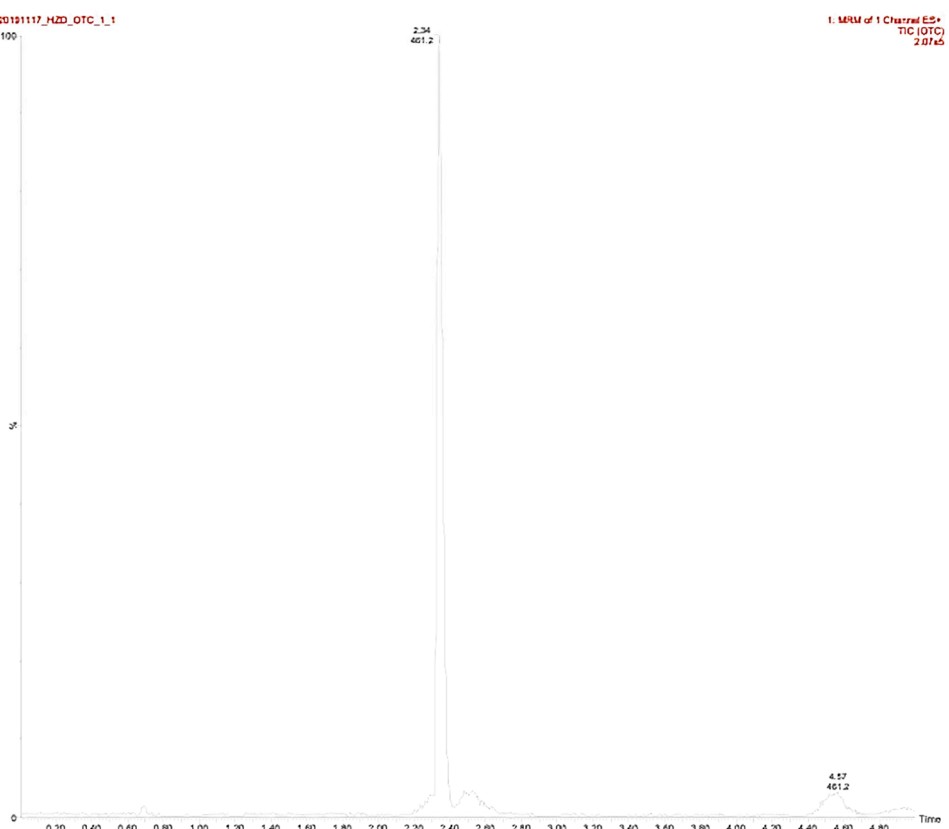

**Figure 2.** Mass spectrometry detection of oxytetracycline content on the first day of group 1.

### 2.3.6. Microbial Community Analysis

The solid matter in the wastewater was centrifuged at 10,000 rpm for 10 min to remove pore water and stored at −80 °C for further analysis. High-throughput sequencing was performed by Majorbio Bio-Pharm Technology Co., Ltd. (Shanghai, China) on the Illumina MiSeq platform. The study used 338F and 806R primers to focus on the V3 and V4 regions of 16S rRNA. The diversity of microbial community structure was analyzed on the free online Majorbio I-Sanger cloud platform (www.i-sanger.com, accessed on 20 June 2020).

## 3. Results and Discussion

### 3.1. Kinetic Degradation Model

In most cases, the degradation pattern of organic pollutants in the environment followed a simple first-order kinetic model, which was described as follows.

$$dC/dt = -kC$$

where C represents the concentration of organic pollutants at a given moment, t represents time, and k is the rate constant. The half-life could then be expressed as [24]:

$$t_{1/2} = \ln 2/k$$

According to the first-order dynamic model,

$$C_t = C_0 \times e^{-kt} \rightarrow C_t/C_0 = e^{-kt} \rightarrow \ln (C_t/C_0) = \ln e^{-kt} = -kt$$

where $C_t$ represents the concentration of organic pollutants at a given moment, $C_0$ represents the initial concentration of organic pollutants, $C_t$, $C_0$, and t are all known parameters in the experiment, and the rate constant k could be obtained by $\ln (C_t/C_0) = -kt$ [25].

Half-life refers to when the concentration of organic pollutants was degraded to half of the initial concentration, representing the degradation speed. Therefore, $C_t = 1/2C_0$, from which $\ln 2 = kt_{1/2}$ could be obtained, meaning that the half-life could be obtained by $t_{1/2} = \ln 2/k$.

It can be seen from Table 4 that all kinds of tetracyclines had $R^2$ values higher than 0.81 in all groups, except for the relatively low $R^2$ in group 2 ($R^2 = 0.69$). (group 2 and group 4 had oxytetracycline added in the middle of the reaction, and therefore the R-square is relatively small). The degradation kinetics of swine wastewater during high-temperature anaerobic digestion were highly consistent with this simple first-order degradation kinetic model, meaning that the reaction conformed to the general law of degradation of organic matter in aqueous solution.

**Table 4.** Fitting results of tetracycline antibiotics for each group in a simple first-order kinetic degradation model.

| Antibiotic | Group | Initial Concentration $C_0$ (ug/L) | Rate Constant k ($d^{-1}$) | Equation | Half-Life $t_{1/2}$ (d) | $R^2$ |
|---|---|---|---|---|---|---|
| OTC | 1 | $3.0518 \pm 0.2328$ | $0.6418 \pm 0.0929$ | $y = 2.6532 \times e^{(-x/1.5580)} + 0.3986$ | $1.0799 \pm 0.1562$ | 0.9644 |
|  | 2 | $4.6182 \pm 1.6674$ | $0.1500 \pm 0.0876$ | $y = 4.5725 \times e^{(-x/6.6670)} + 0.0456$ | $4.6212 \pm 2.6984$ | 0.8940 |
|  | 3 | $3.4847 \pm 0.4583$ | $0.5374 \pm 0.1267$ | $y = 3.1303 \times e^{(-x/1.8607)} + 0.3543$ | $1.2898 \pm 0.3041$ | 0.9095 |
|  | 4 | $3.4199 \pm 1.0734$ | $1.0366 \pm 0.7981$ | $y = 2.3494 \times e^{(-x/0.9647)} + 1.0705$ | $0.9647 \pm 0.7428$ | 0.4217 |
| TC | 1 | $0.8804 \pm 0.0800$ | $0.4569 \pm 0.1057$ | $y = 0.5423 \times e^{(-x/2.1888)} + 0.3381$ | $1.5172 \pm 0.3509$ | 0.9119 |
|  | 2 | $0.8571 \pm 0.0872$ | $1.3607 \pm 0.4106$ | $y = 0.4898 \times e^{(-x/0.7349)} + 0.3673$ | $0.5094 \pm 0.1537$ | 0.8459 |
|  | 3 | $1.0108 \pm 0.1096$ | $0.4890 \pm 0.1282$ | $y = 0.6610 \times e^{(-x/2.0450)} + 0.3499$ | $1.4175 \pm 0.3716$ | 0.8895 |
|  | 4 | $0.8667 \pm 0.1189$ | $0.3506 \pm 0.1169$ | $y = 0.5330 \times e^{(-x/2.8526)} + 0.3337$ | $1.9773 \pm 0.6596$ | 0.8291 |
| CTC | 1 | $7.1269 \pm 0.6777$ | $0.8531 \pm 0.1835$ | $y = 5.2933 \times e^{(-x/1.1723)} + 1.8336$ | $0.8126 \pm 0.1748$ | 0.9224 |
|  | 2 | $5.7384 \pm 0.9615$ | $0.9616 \pm 0.4567$ | $y = 3.4088 \times e^{(-x/1.0400)} + 2.3296$ | $0.7209 \pm 0.3424$ | 0.6919 |
|  | 3 | $7.8822 \pm 1.2341$ | $0.3613 \pm 0.1260$ | $y = 5.3175 \times e^{(-x/2.7678)} + 2.5647$ | $1.9185 \pm 0.6693$ | 0.8153 |
|  | 4 | $6.5028 \pm 1.0580$ | $0.1362 \pm 0.0385$ | $y = 4.7480 \times e^{(-x/7.3450)} + 1.7548$ | $5.0912 \pm 1.4401$ | 0.9293 |
| DOC | 1 | $13.3652 \pm 1.3394$ | $0.8698 \pm 0.2043$ | $y = 9.5854 \times e^{(-x/1.1497)} + 3.7798$ | $0.7969 \pm 0.1872$ | 0.9084 |
|  | 2 | $14.2782 \pm 1.7099$ | $1.0783 \pm 0.3013$ | $y = 10.3165 \times e^{(-x/0.9274)} + 3.9617$ | $0.6428 \pm 0.1796$ | 0.8699 |
|  | 3 | $12.9623 \pm 1.9475$ | $0.3157 \pm 0.0945$ | $y = 9.5809 \times e^{(-x/3.1674)} + 3.3814$ | $2.1955 \pm 0.6568$ | 0.8602 |
|  | 4 | $12.3889 \pm 1.9377$ | $0.2785 \pm 0.0893$ | $y = 8.7491 \times e^{(-x/3.5902)} + 3.6398$ | $2.4885 \pm 0.7980$ | 0.8462 |

The results demonstrated that high-temperature anaerobic digestion had a positive effect on the degradation of tetracycline antibiotics. In this research, except for the fact that the $t_{1/2}$ of chlortetracycline in group 4 was 5 days, the other $t_{1/2}$ values were between 0.5 and 2.4 days. Group 2 had antibiotics added during the reaction, extending its $t_{1/2}$. Composting is widely used as a suitable method for degrading antibiotics in pig manure; however, the $t_{1/2}$ values of compost for degrading OTC, CTC, and TC were 2.66, 16.95, and 22.36 days, respectively. Even if the researchers optimized the degradation kinetic model, the $t_{1/2}$ of the three antibiotics were still 1.14, 8.25, and 10.02 days, respectively [24]. In this study, the high-temperature anaerobic digestion degradation of tetracycline antibiotics in swine wastewater was much shorter than for composting, indicating that high-temperature anaerobic digestion could shorten the overall duration of the reaction.

Adding OTC during the anaerobic reaction made a difference in the degradation process of all TCs, not only OTC. Comparing all $R^2$ data, the largest $R^2$ value for each antibiotic appeared in group 1, which was a control sample with no antibiotic addition and the balanced state in the reactor had not been disrupted. Previous researchers found similar results when fitting the degradation kinetic models of solid and liquid antibiotics [25]. This result could be due to TCs existing in a form that can be dissolved in the liquid phase and adsorbed in the solid phase. Once OTC was exogenously added, it upset the balance of this original state; whenever it was added, it would disturb the dissolution of the liquid phase

and the adsorption of the solid phase, thereby affecting the overall degradation process and finally affecting the degradation kinetics and $R^2$.

### 3.2. Degradation Results of TCs

Group 1 was a control group without an addition of OTC, designed to display the degradation process of TCs under high-temperature anaerobic digestion in the first 20 days without interference from other antibiotics. It could be seen from Table 5 that the removal efficiencies of OTC, CTC, and DOC were higher than 80% (87%, 84%, and 82%, respectively). The removal efficiency of TC in group 1 was the lowest (70%). These results may be due to the initial concentration of tetracycline being lower than OTC, CTC, and DOC in this system.

**Table 5.** TC degradation efficiency in group 1.

| TCs | Initial Concentration $C_0$ (ug/L) | Day 20 Concentration $C_{20}$ (ug/L) | Removal Efficiency |
|---|---|---|---|
| TC | 0.94 | 0.28 | 70.21% |
| OTC | 2.87 | 0.37 | 87.11% |
| CTC | 7.47 | 1.16 | 84.47% |
| DOC | 14.10 | 2.52 | 82.13% |

In research by Wu et al., pilot-scale composting was employed, and CTC, OTC, and TC were degraded 74%, 92%, and 70%, respectively; these results were similar to the degradation results found here [24], though the CTC degradation efficiency in this experiment was much higher. According to a previous study, when anaerobic digestion was adopted, the $t_{1/2}$ of TC and OTC in swine manure samples were 198 days and 62.4 days, respectively, and 5%, 26%, and 88% of the initial TC, OTC, and CTC, respectively, were degraded during 64 days of anaerobic digestion [26]. The CTC degradation result was similar to the experimental results obtained here, though with a much longer digestion period. However, the TC and OTC degradation efficiencies obtained in that study were not ideal. Liu et al.'s experiment focused on the comparison between anaerobic digestion and composting and found that the degradation efficiency of composting was >90%, much better than anaerobic digestion, though it was less efficient [27].

The concentrations of TCs in group 1 dropped rapidly in the first two days, before stabilizing after 10 days (Figure 3). The performance showed that high-temperature anaerobic digestion is suitable for the effective and efficient removal of tetracyclines. Degradation efficiencies for each kind of tetracycline were found to reach levels higher than 70% within 20 days. If the experiment were to be carried out to a complete reaction cycle of 60 days, the removal efficiency of all tetracycline antibiotics would increase further.

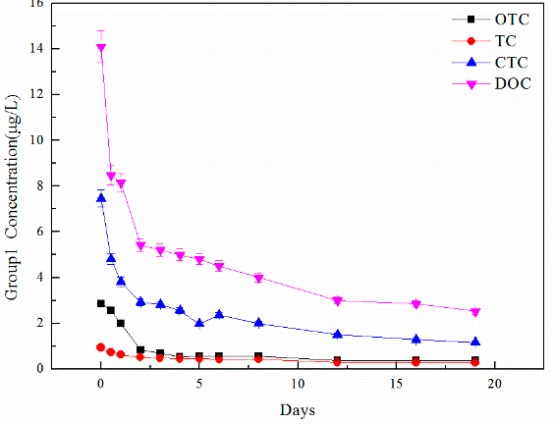

**Figure 3.** Changes in tetracycline content in group 1.

### 3.3. The Effect of Exogenous OTC in Different Time Periods

The removal efficiencies of OTC in the four reactor groups are shown in Table 6. All exhibited good performance (>87%), however, groups 2, 3 and 4 had higher degradation efficiencies than group 1; this indicates that exogenous sources of OTC had a positive impact on the removal efficiency of TCs. A proper concentration of OTC could be digested by the reactor at the same time, further improving the degradation efficiency.

**Table 6.** TC degradation efficiency.

| TCs | Group | Initial Concentration $C_0$ (ug/L) | Adding Concentration (ug/L) | Day 20 Concentration $C_{20}$ (ug/L) | Removal Efficiency |
|---|---|---|---|---|---|
| OTC | 1 | 2.87 | 0.00 | 0.37 | 87.11% |
|  | 2 | 2.66 | 1.00 | 0.45 | 87.70% |
|  | 3 | 3.02 | 1.00 | 0.38 | 90.55% |
|  | 4 | 3.01 | 1.00 | 0.44 | 89.03% |
| TC | 1 | 0.94 | 0.00 | 0.28 | 70.21% |
|  | 2 | 0.88 | 0.00 | 0.29 | 67.05% |
|  | 3 | 1.12 | 0.00 | 0.30 | 73.21% |
|  | 4 | 1.00 | 0.00 | 0.28 | 72.00% |
| CTC | 1 | 7.47 | 0.00 | 1.16 | 84.47% |
|  | 2 | 6.12 | 0.00 | 1.53 | 75.00% |
|  | 3 | 9.29 | 0.00 | 2.23 | 76.00% |
|  | 4 | 7.15 | 0.00 | 2.10 | 70.63% |
| DOC | 1 | 14.10 | 0.00 | 2.52 | 82.13% |
|  | 2 | 14.82 | 0.00 | 2.30 | 84.46% |
|  | 3 | 15.13 | 0.00 | 3.00 | 80.17% |
|  | 4 | 14.53 | 0.00 | 2.87 | 80.25% |

With respect to the degree of degradability, additional OTC could promote degradation. Over time, group 3 had the best removal performance (90.55%). The concentrations of OTC in groups 2 and 4 increased suddenly, accounting for the exogenous addition; this was not observed in group 3 (Figure 4). Taking the addition method into consideration, multiple additions of small amounts would likely result in a better performance, lowering the degradation load of the reactor. Comparing group 2 with group 4, the performance gained by addition on day 4 was better, as most of initial OTC was degraded so that the exogenous OTC could be fully degraded as well. This result supported the lower load theory.

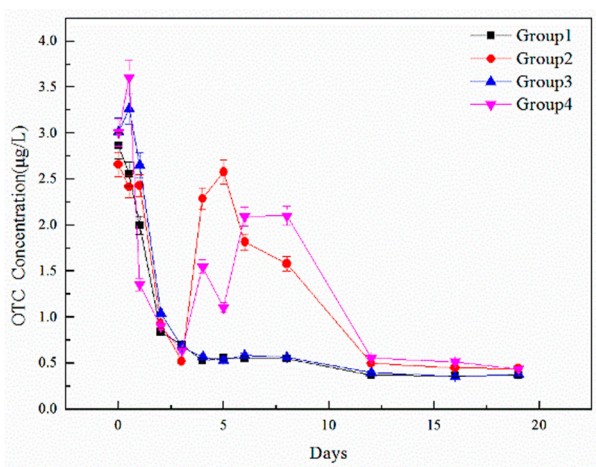

**Figure 4.** Changes in oxytetracycline content.

As shown in Table 6, TC degradation in groups 2, 3 and 4 was 67%, 73%, and 72%, respectively; the DOC for these three groups was 84%, 80%, and 80%, respectively. From the results, exogenous OTC did not appear to affect the degradation of TC and DOC. Importantly, attention should be paid to the efficiency of CTC degradation, which dropped from 84% to about 70% in the other groups. A possible reason for this is that OTC and CTC have similar molecular structures, indicating similar chemical natures. Therefore, if exogenous OTC were added, some CTC reactions might be seized by the addition as enzymes begin to interact with the OTC as well.

### 3.4. The Effect of OTC Addition on Heavy Metals

Metal ions had various effects on the anaerobic digestion process. The heavy metals in the liquid phase of the four reactor groups are shown in Figure 5. It was seen that the concentration of Cu was highest, followed by Fe and Zn. From experimental day 1 to day 2, the heavy metal concentrations dropped; Cu in group 1 decreased from 10.97 mg/L on day 1 to 2.41 mg/L on day 2. The largest decrease in Cu concentration was in group 4, which decreased from 20.18 mg/L on day 0.5 to being undetectable in the liquid phase by day 5. A possible reason for this phenomenon was that anaerobic digestion produces large amounts of organic acids after entering the acid production stage. Some scholars previously digested swine wastewater in combination with food waste and found that trace metals (such as Co, Mo, Ni, and Fe) in swine wastewater could improve biogas production and enhance digestion stability [28]. These results are consistent with the conclusion drawn by Zhou et al. [29]. Other researchers have found significant positive correlations between some antibiotic resistance genes (ARG) and the concentrations of copper, zinc, lead, cobalt, and mercury [30,31]. Organic acids are a type of dissolved organic matter (DOM), and functional groups (such as amide, carboxyl, phenol, and hydroxyl groups) could effectively bind heavy metals through chelation or complexation [32]. In this experiment, the carboxyl groups in the large quantities of organic acids produced would be expected to chelate and complex the heavy metals, accounting for the decreasing Cu concentration.

The influence of the addition of OTC on heavy metals in wastewater can also be studied by changes in content. On the fourth day, OTC was added to group 2, and the content of heavy metals suddenly decreased, showing a completely opposite trend to what was expected based on the behavior of group 1 (Figure 5a,b). This demonstrated that it was likely that the functional groups in OTC could effectively bind heavy metals in a similar manner to organic acids [32]; the same phenomenon is also shown in the comparison between group 1 and group 4 (Figure 5a,d). The change in Cu content was more obvious than that of the other metals not only because Cu was present in a higher concentration but also because Cu is known to have a high complex affinity as a ligand with tetracycline antibiotics [33].

This study demonstrated that high concentrations of Zn and tetracycline may lead to a decrease in total nitrogen concentration, which confirmed results from Fan et al.'s research which showed that Zn and tetracycline inhibit anaerobic ammonia oxidation performance [34]. The Zn concentration also correlated with the total antibiotic concentration, as well as with several specific antibiotics, such as oxytetracycline ($p < 0.01$). Zinc is known to bind several antibiotics, including quinolones, tetracyclines, and macrolides [35], explaining the observed correlation.

There was also a high correlation between changes in pH and metal concentrations in group 1 (Figures 5 and 6). In anaerobic digestion, the change of pH mainly results from the accumulation of organic acid; therefore, the observed reduction in heavy metals was likely due to production of this acid. Together, these results confirmed that DOM is capable of reducing the concentration of heavy metals [36].

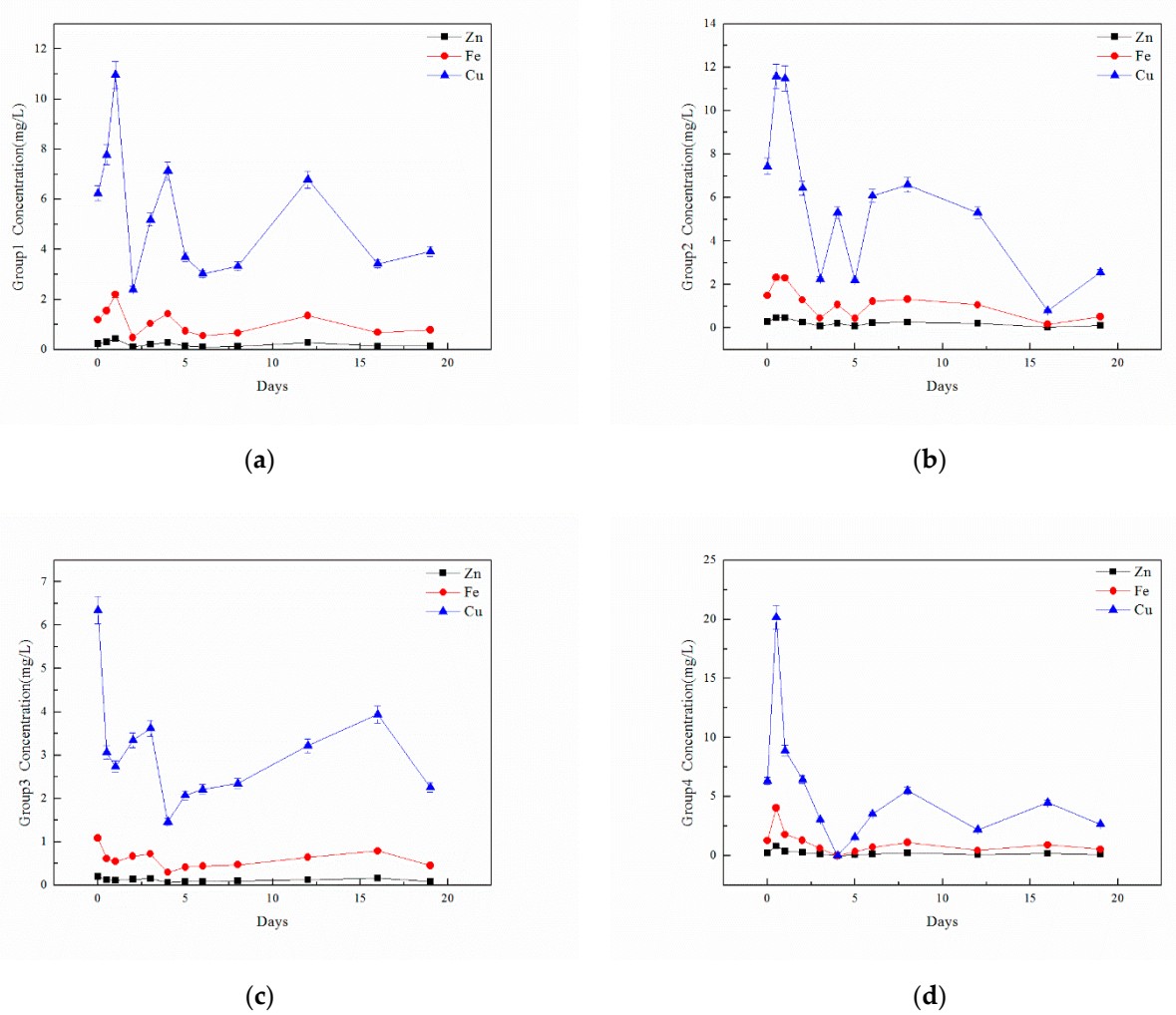

(**a**)  (**b**)

(**c**)  (**d**)

**Figure 5.** The concentrations of Zn, Fe, and Cu in the liquid phase of group 1 (**a**), group 2 (**b**), group 3 (**c**), and group 4 (**d**).

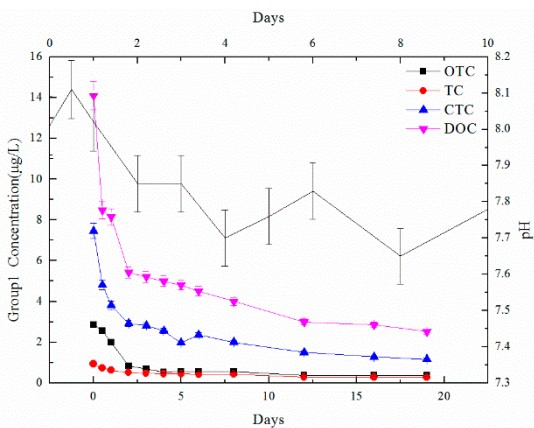

**Figure 6.** Changes in TC content and pH in group 1.

Overall, metal ions and antibiotics affected the reaction process mainly through coordination into complexes, as did the exogenous addition of OTC. In addition, the organic acids produced during the anaerobic reaction phase were also able to affect the heavy metal content of the wastewater.

### 3.5. The Effects of Adding OTC on Microbial Community

In anaerobic processes, microorganisms convert complex organic compounds into simple, chemically stable compounds [37]. The most abundant bacterium identified in the swine manure was the hydrolytic fermentative *Clostridium* genus. These microbes could tolerate the natural concentration of the substrate, as well as high concentrations of organic acids and alcohols. The second most dominant genus was *Bacilli*, with the ability to produce acetate and lactic acid [38]. Most archaea found were considered to be methanogens [39]. Other microorganisms that may be present during anaerobic digestion also include *Gammaproteobacteria* [40], in addition to many *Deltaproteobacteria* that have been identified as vegetative acetogenic bacteria causing acetic acid production [41]; according to their fermentation capacity, these can be divided further into *Bacteroidia*, *Actinobacteria*, and *Anaerolineae* [39].

In this study, the most abundant bacteria during the initiation of the experiment were *Clostridia*, *Pseudomonadales*, and *Erysipelotrichales* (Figure 7). After the reaction started, *Pseudomonadales* and *Erysipelothrix* were dramatically reduced. These two microbes are mainly aerobic bacteria and have difficulty growing in an anaerobic state, which likely explains their reduced numbers. It was observed that the composition of groups 1 and 3 was very similar (Figure 7); the difference between them was that the proportion of *Flavobacteriales* in group 3 on day 0 was significantly higher than that found in group 1. From the overall digestion process in group 1, *MBA03* bacteria appeared to have been growing during the whole reaction; on day 15, the *MBA03* abundance ratio had reached about 10%. The same results were observed in group 3; on day 15, abundance reached approximately 25%. There was about a 30% abundance ratio in group 2 until day 4, when, after adding OTC, the proportion of this bacteria decreased. This is possibly because the addition of OTC inhibits the activity of this strain, which is why this group showed a different trend to the others. Comparing the microbial content of the four groups on day 15, group 3 had a higher amount of *D8A-2*, while *Rhodocyclales* was more abundant in group 1.

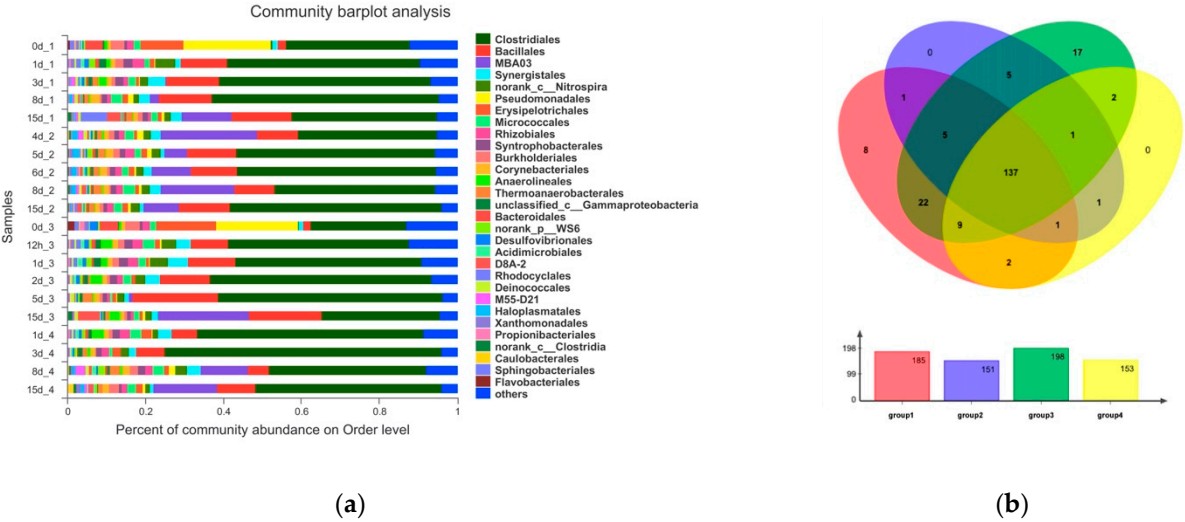

(**a**)　　　　　　　　　　　　　　(**b**)

**Figure 7.** (**a**) Microbial composition at the order level. (**b**) The number of microorganisms at the order level.

The distribution of species is represented with Venn diagrams (Figure 7b) that show the composition similarity and overlap from the four reactors [42,43]. There were 8 (group 1), 0 (group 2), 17 (group 3) and 0 (group 4) unique species observed in each group, indicating that adding OTC during the digestion kills all unique bacterial strains. Conversely, if OTC was added at the beginning of the reaction, it appeared to increase biodiversity; this may be because some anaerobic bacteria that grow during the reaction may be inhibited by OTC, and therefore more drastically affected if it is added during the digestion process.

The bacteria with the highest proportions in the anaerobic reaction systems were *Clostridia*, *Bacilli*, and *Actinobacteria* (Figure 7a), which is consistent with previous research [22]. Some studies found that the most abundant microorganisms in the biological filtration systems for the treatment of swine wastewater were *Chlorellaceae*, *Chlamydomonaceae*, and *Bacillusaceae* [44]. *Bacilli* ranked second in abundance during the whole reaction period in this study, and its abundance continued to increase with degradation time. In this experiment, the fastest increase in *Bacilli* was observed in group 3; from the beginning of the experiment to day 5, its growth in proportion was steady, which may be related to the degradation of organic matter or heterotrophic nitrification.

The exogenous addition of OTC was found to inhibit the activity of some microbes and reduced the proportion of certain dominant microbes (such as *MBA03*), while resulting in the death of certain others. This was most obvious when OTC was added midway through the reaction.

### 3.6. Interaction with TCs

Microorganisms played an important role in the geochemical cycling of metals. These metal cycles were driven by microorganisms, as metals are essential for microbial nutrition (Fe, Co, Cu, Ni, and Zn in particular) [45].

### 3.6.1. Relationship between TCs, Heavy Metals and Microorganisms

Former research found that *Pseudomonadales* secreted a large amount of HCN in the early stages of growth, which formed complexes with heavy metals to promote the dissolution and migration of heavy metals. From Figure 7a, the proportion of *Pseudomonas* significantly decreased in the initial stage of degradation, likely mainly due to the anaerobic environment. In addition, in group 3 (Figure 5c), at the beginning of the reaction, the concentration of metal ions in the liquid phase dropped significantly. The possible reason for this is that HCN secreted by *Pseudomonas* and heavy metals form complexes that promote its dissolution and migration. In this experiment, the abundance of *Pseudomonas* decreased, and the concentration of heavy metals in the liquid phase also decreased; therefore, the comprehensive pollution of heavy metals and antibiotics may exert complicated effects on environmental organisms. Taking Zn as an example, Zn may affect the effectiveness of antibiotics on organisms; it is known that Zn combined with TC can reduce its effectiveness [46], while for most quinolone drugs, zinc can enhance efficacy [35]. Some reports have confirmed the synergistic effect of Zn and quinolones on the biofilm of *Pseudomonas aeruginosa* and the enhancement of drug resistance in *Acinetobacter baumannii* [47,48]. It is also known that antibiotics and Zn have a synergistic and selective effect on bacteria in the environment, as antibiotics in the environment apply a common selective pressure for bacteria, and high concentrations of heavy metals may kill bacteria [49].

### 3.6.2. Relationship between TCs, pH, and Microorganisms

A previous experimental study has shown that in the anaerobic operation system, the abundance of *Nitrospira* decreased due to a reduction of dissolved oxygen and a relatively high pH value, resulting in a loss of growth advantage [50]; this phenomenon was observed in this experiment as well. As shown in Figures 6 and 8a, *Nitrospira* in group 1 quickly gained the advantage on the first day, but because of the reduction of dissolved oxygen during anaerobic conditions and a relatively high pH value its abundance later decreased.

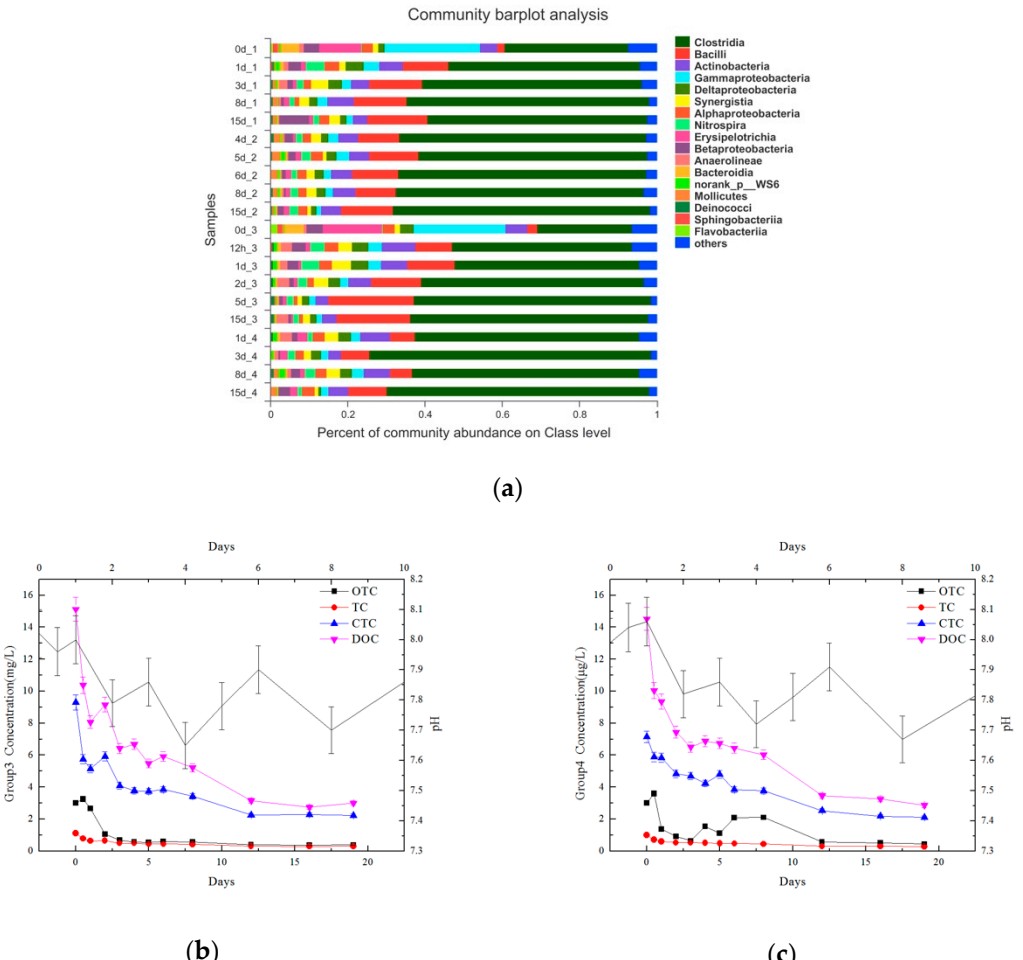

**Figure 8.** (**a**) Microbial composition at the class level. (**b**) Changes of antibiotic content and pH in group 3. (**c**) Changes in antibiotic content and pH in group 4.

Generally speaking, the most abundant bacteria in pig manure are the hydrolytic fermentative species of *Clostridium*, probably due to the fact that these bacteria can use a wide range of substrates and tolerate high concentrations of organic acids and alcohols. The second most abundant are *Bacilli* [38], as they possess the ability to produce acetate and lactic acid. Other microorganisms may also exist in the anaerobic digestion process, including *Gammaproteobacteria* [40]. Many *Deltaproteobacterias* [41] that had been identified as vegetative acetogens might also be present and cause acetogenesis. According to their fermentation ability, they can be divided into *Bacteroidia*, *Actinobacteria*, *Anaerolineae*, and others [39]. These organisms were also found in this study; taking group 4 as an example (Figure 8a,c), the proportion of *Gammaproteobacteria* decreased rapidly at the beginning of the reaction before stabilizing. This is likely due to the fact that *Gammaproteobacteria* consume oxygen and the anaerobic conditions later in the reaction were insufficient to support its survival. A similar trend occurred in groups 1, 2, and 3, with the proportion of *Gammaproteobacteria* continuing to decrease. At the beginning of the reaction, the abundance of Deltaproteobacteria increased clearly in groups 1 and 3 between the comparisons on day 0 and on the first day.

In summary, the effect of heavy metals on the digestion process was mainly through forming coordination bonds with antibiotics and some reaction byproducts. Heavy metals also had a toxic effect on some microorganisms. Microorganisms were mainly affected by pH fluctuations caused by the accumulation of volatile fatty acids, which controlled the process of anaerobic digestion.

## 4. Conclusions

High-temperature anaerobic digestion effectively removed TCs in swine wastewater. All tetracycline antibiotics had a removal efficiency of more than 70% within the 20-day reaction cycle. The addition of OTC during the reaction process affected the first-order reaction kinetics of the antibiotics, and exogenous oxytetracycline inhibited the degradation process of chlortetracycline, though it had no effect on tetracycline or doxycycline. The exogenous addition of oxytetracycline also mainly affected the content of heavy metals by coordinating complex formation. It also inhibited the activity of some microorganisms, reducing the proportion of dominant microbes, such as *MBA03*, and killing some other specific strains. The microorganisms in the anaerobic digestion process were greatly affected by pH, and the abundance of *Nitrospira* was found to decrease with increasing pH value.

**Author Contributions:** Conceptualization, Z.H.; methodology, Z.H.; software, Z.H.; validation, Z.F. and M.Z.; formal analysis, Z.H. and R.K.; investigation, N.W. and Q.S.; resources, Z.H. and Z.F.; data curation, Z.F.; writing—original draft preparation, Z.H.; writing—review and editing, Q.S. and X.L.; visualization, Z.H. All authors have read and agreed to the published version of the manuscript.

**Funding:** This research received no external funding.

**Institutional Review Board Statement:** Not applicable.

**Informed Consent Statement:** Not applicable.

**Data Availability Statement:** Not applicable.

**Acknowledgments:** The authors would like to thank the National Major Science and Technology Program for Water Pollution Control and Treatment (Grant No. 2017ZX07602-001) and ABA Chemicals for their support and donations.

**Conflicts of Interest:** The authors declare no conflict of interest.

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
