# Peer review of "The Effect of Exogenous Oxytetracycline on High-Temperature Anaerobic Digestion of Elements in Swine Wastewater"

_water, doi:10.3390/w13243497_

Round 1

Reviewer 1 Report

Some specific comments to the authors are:

Introduction

Page 1 line 29. Give the wastewater production in terms of m3

Page 2 line 64. Mention examples of good removal of antibiotics in anaerobic digestion systems, for example the citations Garcia-Sanchez et al., 2016 WST among others 2 line 72-75.  Puede dar una referencia para soportar esta idea.

Page 2 lines 78-80.  … There were few researches aiming to the effect of antibiotic’s content on high temperature digestion and how various subjects in wastewater affect each other in the process. The wording of these lines is confusing. Write again.

Page 3 lines 87-88. Explain in more detail.

Page 3, lines 88-89. Please check the English of the sentence (the verb tense).

Materials and Methods

Page 3, lines 95 -96. Please define which microorganisms and provide a reference or delete the phrase as it is very general

Page 3, lines 97 -101. The information is wrongly located put it in an appropriate section

Page 3, lines 104 -106. Information not necessary in this section

Page 3, line 113 Please define the amount of granular sludge used in each digester.

Page 3, line 116 the next sentence is not understood: 1 ug OTC was added in group 2,3,4 in different methods

Page 3, lines 115 a 119. It would be better to put the information of these lines in a table instead of in the text

Page 3, lines 122 a 138. The writing is very confusing. Separate ideas

Page 4, lines 134 a 135. Please mention what methodology was used for the identification of microorganisms and put a reference

Author Response

The answers for Reviewer 1

Introduction

Page 1 line 29. Give the wastewater production in terms of m3

A: Modified in manuscript.

Page 2 line 64. Mention examples of good removal of antibiotics in anaerobic digestion systems, for example the citations Garcia-Sanchez et al., 2016 WST among others 2 line 72-75.  Puede dar una referencia para soportar esta idea.

A: Sorry, I don’t understand the meaning of last sentence, just interpreted by translator. Some references were added in introduction.

Researchers evaluated the impact of different concentrations of tetracycline on the performance of anaerobic treatment, and found no major sustained impact on methane production, demonstrating that anaerobic digestion was stable and reliable in the presence of tetracycline [14]. During the anaerobic digestion of manure from medicated calves, the chlortetracycline (CTC) concentration decreased approximately 75% during the 33 day digestion period [15]. Wang et al.'s results showed that the removal rate of TCs in the supernatant could be up to 90%-100% during swine manure anaerobic digestion. However, their removal rates in the solid phase were only around 40% [16]. The OTC removal efficiency in a different thermophilic composting study was found to be over 90% [17].

Page 2 lines 78-80.  … There were few researches aiming to the effect of antibiotic’s content on high temperature digestion and how various subjects in wastewater affect each other in the process. The wording of these lines is confusing. Write again.

A: The new sentence is as followed.

As the content of antibiotics varies in different swine wastewater, there are few studies aiming to examine the effect of antibiotic content on high temperature swine wastewater digestion, and how various components in the wastewater affect each other in the process.

Page 3 lines 87-88. Explain in more detail.

A: Pig manure degrades easily in the environment. Therefore, fresh pig manure is original and primitive, which is suitable for anerobic digestion.

About the components of pig manure, its solid content is 5%, and wastewater basic indicators are shown in Table 1. Initial concentrations of TCs are shown in Table 5 and Figure 3. The contents of heavy metals are shown in Figure 5.a. The original microorganism is shown in Figure 7 labeled by 0. The reason for not summarizing at the beginning of the article is to prevent duplication of information and increase the readability of the article.

Page 3, lines 88-89. Please check the English of the sentence (the verb tense).

A: The new sentence is as followed.

The solid content of pig manure was about 5%, and the basic indicators such as pH, total organic carbon (TOC), total phosphorus (TP), and total nitrogen (TN) are shown in Table 1.

Materials and Methods

Page 3, lines 95 -96. Please define which microorganisms and provide a reference or delete the phrase as it is very general

A: Deleted

Page 3, lines 97 -101. The information is wrongly located put it in an appropriate section

A: This part has been moved to 2.3.2. Solid phase extraction. I think 2.3.2 is about TCs solid phase extraction, which should introduce antibiotic information.

Page 3, lines 104 -106. Information not necessary in this section

A: This part has been deleted.

Page 3, line 113 Please define the amount of granular sludge used in each digester.

A: ‘quantitative’ has been changed to ‘50 grams of solidified granular sludge’.

Page 3, line 116 the next sentence is not understood: 1 ug OTC was added in group 2,3,4 in different methods

A: After the digestion was started, groups 2, 3, and 4 were spiked with 1 µg of OTC by different methods.

Page 3, lines 115 a 119. It would be better to put the information of these lines in a table instead of in the text.

A: A table has been added in this part.

Group

0

2

4

6

8

10

12

14

16

18

20

1

0

0

0

0

0

0

0

0

0

0

0

2

1.00

0

0

0

0

0

0

0

0

0

0

3

0.33

0

0.33

0

0.33

0

0

0

0

0

0

4

0

0

1.00

0

0

0

0

0

0

0

0

Page 3, lines 122 a 138. The writing is very confusing. Separate ideas

A: This part aimed to introduce some details of the structure of the research. I separated this part into two paragraphs. The first one included the reason why we chose first 20 days and the second stated the analysis dimensions. I also changed some expressions and added sampling timings.

Page 4, lines 134 a 135. Please mention what methodology was used for the identification of microorganisms and put a reference

A: Added a new part: 2.3.6. Microbial community analysis

The solid matter in the wastewater was centrifuged at 10,000 rpm for 10 minutes to remove pore water and stored at -80°C for further analysis. High-throughput sequencing was performed by Majorbio Bio-Pharm Technology Co., Ltd. (Shanghai, China) on the Illumina MiSeq platform. The study used 338F and 806R primers to focus on the V3 and V4 regions of 16S rRNA. The diversity of microbial community structure was analyzed on the free online platform of the Majorbio I-Sanger cloud platform (www.i-sanger.com).

Reviewer 2 Report

In this work, the influence of oxytetracycline on the high temperature anaerobic digestion is explored, through its addition at different times (fed-batch). The results are interesting and may be useful for the scientific community that studies the anaerobic digestion process of pig wastewater. However, before the manuscript is accepted for publication, the authors should attend to the following:

a) The motivation and scope of work are well established in the introduction. However, to date different works have already been reported where anaerobic digestion (under mesophilic and thermophilic conditions) in the presence of different types of tetracyclines is studied. Therefore, I suggest that an exhaustive review of the literature be carried out and a broader discussion on the contribution of this work be presented in the introduction.

b) More information on the composition of the pig wastewater used for the study should be included, in particular the content of the different antibiotics present. Likewise, the adverse consequences to the environment should also be discussed, if the antibiotics present in the pig wastewater are not eliminated.

c) The results could be presented using response surfaces, which would favor the understanding between the interactions of the studied factors.

d) The authors establish that the exogenous addition of OTC affects degradation tetracycline antibiotics and assume that the kinetic degradation follows a first-order model. Anaerobic digestion is a complex process so a first-order model can be very limited. A discussion of more complete models that can better describe degradation kinetics would be desirable.

e) The manuscript exhibits some grammatical errors and misspellings. It can certainly benefit from an extensive language revision.

Author Response

The answers for Reviewer 2

In this work, the influence of oxytetracycline on the high temperature anaerobic digestion is explored, through its addition at different times (fed-batch). The results are interesting and may be useful for the scientific community that studies the anaerobic digestion process of pig wastewater. However, before the manuscript is accepted for publication, the authors should attend to the following:

  1. a) The motivation and scope of work are well established in the introduction. However, to date different works have already been reported where anaerobic digestion (under mesophilic and thermophilic conditions) in the presence of different types of tetracyclines is studied. Therefore, I suggest that an exhaustive review of the literature be carried out and a broader discussion on the contribution of this work be presented in the introduction.

A: Some references were added in introduction.

Researchers evaluated the impact of different concentrations of tetracycline on the performance of anaerobic treatment, and found no major sustained impact on methane production, demonstrating that anaerobic digestion was stable and reliable in the presence of tetracycline [14]. During the anaerobic digestion of manure from medicated calves, the chlortetracycline (CTC) concentration decreased approximately 75% during the 33 day digestion period [15]. Wang et al.'s results showed that the removal rate of TCs in the supernatant could be up to 90%-100% during swine manure anaerobic digestion. However, their removal rates in the solid phase were only around 40% [16]. The OTC removal efficiency in a different thermophilic composting study was found to be over 90% [17].

  1. b) More information on the composition of the pig wastewater used for the study should be included, in particular the content of the different antibiotics present. Likewise, the adverse consequences to the environment should also be discussed, if the antibiotics present in the pig wastewater are not eliminated.

A: About the components of pig manure, its solid content is 5%, and wastewater basic indicators are shown in Table 1. Initial concentrations of TCs are shown in Table 5 and Figure 3. The contents of heavy metals are shown in Figure 5.a. The original microorganism is shown in Figure 7 labeled by 0. The reason for not summarizing at the beginning of the article is to prevent duplication of information and increase the readability of the article.

Just as stated in the article, TCs is the most important in livestock. This is the reason why TCs is the target of the research. OTC is an example of TCs. If all antibiotics were tested, it would not only increase the cost of the experiment, but also make the research messy.

About adverse consequences to the environment, more references have been added in introduction.

Antibiotics are difficult to degrade in the natural environment, and can have a terrible influence on soil and water. Antibiotic administration has direct effects on the enteric and pathogenic bacteria in livestock animals, and environmental bacteria also are influenced if antibiotic residues and metabolites are released into the environment [9]. Currently, antibiotic-resistant bacteria with resistance to sulfonamides and tetracyclines are detected ubiquitously in Asia [10].

  1. c) The results could be presented using response surfaces, which would favor the understanding between the interactions of the studied factors.

A: Thank you very much for your comments, and I personally think that response surface analysis is very appropriate. Because I didn’t know much about this aspect before, I went to study it specially to enrich the content of my article and provide a more reliable argument for the article. But when preparing to analyze, I discovered that when I was saving microbial data, the specific values and percentages were not stored. It was just that some heat maps and histograms were saved on the analysis platform at that time. The microbiological testing and analysis were done by an online microbiological company. I have completed the testing for more than a year. The company only keeps data for three months for customers. I specifically contacted the company, and indeed there is no data backup. Microbes are an important transmission link between antibiotics, heavy metals and pH, and correlation analysis is mainly done around microbes. If there is no microbiological data, the results of response surface analysis will not be particularly significant. Therefore, after careful consideration, I have stated this fact to you, and I hope you can understand. I will be careful in future experiments.

  1. d) The authors establish that the exogenous addition of OTC affects degradation tetracycline antibiotics and assume that the kinetic degradation follows a first-order model. Anaerobic digestion is a complex process so a first-order model can be very limited. A discussion of more complete models that can better describe degradation kinetics would be desirable.

A: Just as the reviewer mentioned, anaerobic digestion is a complicated process, which is difficult to describe by degradation kinetics. There is no doubt that more complete models are better to describe the degradation process. However, degradation kinetic was not the core point in the research, and this part was employed to reproduce the degradation process of antibiotics, so that readers could understand better. So first-order model is enough to finish this assignment. Some researchers in the same field used first-order model as well, such as Wang, R.; Feng, F.; Chai, Y.F.; Meng, X.S.; Sui, Q.W.; Chen, M.X.; Wei, Y.S.; Qi, K.M. Screening and quantitation of residual antibiotics in two different swine wastewater treatment systems during warm and cold seasons. Science of the Total Environment 2019, 660, 1542-1554.

  1. e) The manuscript exhibits some grammatical errors and misspellings. It can certainly benefit from an extensive language revision.

A: Thanks for your advice. The manuscript has finished English editing service.

Reviewer 3 Report

Please find attached my remarks

Author Response

The answers for Reviewer 3

In the present work Hu et al. showed that high temperature anaerobic digestion is more suitable for the removal of oxytetracycline.

A: Thanks

The title is good however the abstract although accurate could be improved into a more fluent form.

A: The abstract has been modified into a more fluent form.

Tetracycline antibiotics (TCs) are a common antibiotic type found in swine wastewater. Oxytetracycline (OTC) is a significant type of TCs. This study mainly explored the influence of OTC on high temperature anaerobic digestion by adding OTC at different times during digestion. The results showed that high temperature anaerobic digestion was suitable for the removal of TCs, with 87% OTC removal efficiency by the 20th day. Additionally, OTC added from external sources was found to inhibit the chlortetracycline degradation process and affect the first-order degradation kinetic model of TCs. A complexation reaction was the main approach by which OTC affected the heavy metal content of the water. Exogenous addition of OTC was found to inhibit the activity of some digester microbial strains, reduce the proportion of dominant strains such as MBA03, and kill certain specific strains. This performance alteration was most obvious when OTC was added in the middle of the reaction.

There several language issues, mainly in regards to syntax.

A: Thanks for your advice. The manuscript has finished English editing service.

The introduction is well structured and cited

A: Thanks

Materials and methods

2.1. how were the measurements of table 1 made?

A: I need to say thanks to the reviewer. This is a mistake I made. I put the diluted results of TOC and TN, and wrong result of TP, in the article. The correct results have been modified in Table 1. Thanks again and I will pay more attention to details in later research.

The basic indicators were tested after mixing, centrifuging and diluting. TOC and TN were detected by the TOC analyzer, and TP was analyzed in molybdate calibration method manually. This part has been added in 2.1. Swine wastewater and granular sludge.

Table. Basic indicators of swine wastewater.

Indicator

Value (unit)

pH

8.01±0.1

TOC

2598.8±130.0 mg/L

TN

1374.0±54.4 mg/L

TP

112.8±5.3 mg/L

2.3.1 ultra-high performance liquid chromatography or "ultra-performance liquid chromatography"? uhplc/uplc?

A: It’s uplc, ultra-performance liquid chromatography. The typo has been corrected in manuscript.

2.3.3 what was the Data Rate/Acquisition Frequency (Hz)

A: There was 12 samples taken from the digestion process. The sampling timings were 0, 0.5, 1, 2, 3, 4, 5, 6, 8, 12, 16, 19 days. The data acquisition frequency has been added in 2.2. Experimental design.

Line 445-446: state the reason why

A: OTC will increase the relative content of acetogenic bacteria, Aspartame, and promote the metabolism of more organic acids to produce volatile fatty acid VFAs, leading to an increase in the concentration of VFAs, which in turn leads to a decrease in ph. Conversely, the excessive accumulation of VFAs will also inhibit the entire process.

This research has been finished by my coauthor, Zijing Fan, and published in another article. The article will not include more details about this part. The reference is as followed.

Fan, Z.; Zhang, M.; Chen, X.; Hu, Z.; Shu, Q.; Jing, C.; Luo, X., Effects of Lighter Dose of Oxytetracycline on the Accumulation and Degradation of Volatile Fatty Acids in the Process of Thermophilic Anaerobic Digestion of Swine Manure. Sustainability 2021.

This manuscript is a resubmission of an earlier submission. The following is a list of the peer review reports and author responses from that submission.